# Implementation of a Patient Questionnaire in Community Pharmacies to Improve Care for Patients Using Combined Antithrombotic Therapy: A Qualitative Study

**DOI:** 10.3390/pharmacy11030080

**Published:** 2023-04-27

**Authors:** Renate C. A. E. van Uden, Marit A. Bakker, Stephan G. L. Joosten, Karina Meijer, Patricia M. L. A. van den Bemt, Matthijs L. Becker, Marcia Vervloet

**Affiliations:** 1Pharmacy Foundation of Haarlem Hospitals, Boerhaavelaan 24, 2035 RC Haarlem, The Netherlands; 2Department of Clinical Pharmacy, Spaarne Gasthuis Hospital, Boerhaavelaan 22, 2035 RC Haarlem, The Netherlands; 3Department of Clinical Pharmacy and Pharmacology, University Medical Center Groningen, University of Groningen, P.O. Box 30.001, 9700 RB Groningen, The Netherlands; 4Community Pharmacy BENU Pharmacy Nieuwpoort, Jan van der Heydenweg 352, 3401 RJ IJsselstein, The Netherlands; 5Department of Haematology, University Medical Center Groningen, University of Groningen, P.O. Box 30.001, 9700 RB Groningen, The Netherlands; 6Nivel, Netherlands Institute for Health Services Research, P.O. Box 1568, 3500 BN Utrecht, The Netherlands

**Keywords:** implementation research, guideline adherence, antithrombotic therapy, qualitative study, primary care, community pharmacy

## Abstract

For several indications or combinations of indications the use of more than one antithrombotic agent is required. The duration of combined antithrombotic therapy depends on indication and patient characteristics. This study investigated the use of an antithrombotic questionnaire tool that had been developed for pharmacists to detect patients with possible incorrect combined antithrombotic therapy. The objective of this study was to identify potential barriers and facilitators that could influence the implementation of the developed antithrombotic questionnaire tool in daily community pharmacy practice. A qualitative study was conducted at 10 Dutch community pharmacies in which the antithrombotic questionnaire tool had been used with 82 patients. Semi-structured interviews were conducted with pharmacy staff who used the antithrombotic questionnaire tool. The interview questions to identify barriers and facilitators were based on the Consolidated Framework for Implementation Research. The interview data were analysed using a deductive thematic analysis. Ten staff members from nine different pharmacies were interviewed. Facilitators for implementation were that the questionnaire was easily adaptable and easy to use, as well as the relative short duration to administer the questionnaire. A possible barrier for using the questionnaire was a lower priority for using the questionnaire at moments when the workload was high. The pharmacists estimated that the questionnaire could be used for 70–80% of the patient population and they thought that it was a useful addition to regular medication surveillance. The antithrombotic questionnaire tool can be easily implemented in pharmacy practice. To implement the tool, the focus should be on integrating its use into daily activities. Pharmacists can use this tool in addition to regular medication surveillance to improve medication safety in patients who use combined antithrombotic therapy.

## 1. Introduction

Antithrombotic therapy is the cornerstone for primary and secondary prevention of both arterial and venous thrombosis. For several indications the use of more than one antithrombotic agent is required. Combined antithrombotic therapy carries a two- to four-fold increased bleeding risk compared to monotherapy [1,2]. For most indications, the risk of a new thrombotic event decreases over time, while the bleeding risk does not decrease [3,4,5]. Therefore, the guidelines recommend that combined antithrombotic therapy should only be used for a limited period of time and not indefinitely [6,7,8,9,10,11,12,13]. The duration depends on the indication and patient characteristics. Considering the high bleeding risk in combined antithrombotic therapy it is important that combined antithrombotic therapy is stopped at the right moment.

Several studies have shown that a considerable proportion of patients who used combined antithrombotic therapy had no clear indication for the therapy (anymore). These patients were exposed to an unnecessarily high bleeding risk [14,15,16,17]. Although physicians can choose to deviate intentionally from the guideline, for instance, when a patient has a high ischemic risk after multiple thromboembolic events, most deviations have been unintentional when treatment was continued after the intended stop date [14,15]. In previous studies, we identified that 13.7% of patients who used dual antiplatelet therapy (DAPT) prior to hospital admission and 27.9% of admitted patients who used dual or triple antithrombotic therapy had an unintentional guideline deviation [14,15]. This suggests that the current medication surveillance on combined antithrombotic therapy is not adequate. When pharmacists informed the physician of the possible unintentional deviation and advised the physician on how to adjust the antithrombotic therapy, the acceptance rate of the given advice was 100% and 90.2%, respectively [14,15]. Thus, pharmacists can play an important role in detecting and correcting potentially incorrect antithrombotic therapy.

In order to assess whether combined antithrombotic therapy is correct, a pharmacist needs information on indication and the start date of the antithrombotic therapy. However, community pharmacists often lack this information. Our previous study showed that patients often had adequate knowledge of why they were using antithrombotic therapy (indication) and when the therapy had been started (start date). To obtain this information from patients, we developed an antithrombotic questionnaire tool [18]. The tool consists of nine questions that ask patients for the indication for the prescribed therapy (presented as a list of indications to be checked yes or no), the start date of the therapy, and the intended duration of the antithrombotic therapy (Appendix A). Patients can also indicate that they do not know the answer. In our previous study, a pharmacist asked the questions and registered the answers. The majority of the patients knew both the indication and the start date of their antithrombotic therapy, but not the intended duration. Information provided by patients was correct for 98% of the patients who could answer the questions on indication and start date compared to information in the medical record, enabling the pharmacist to contact the prescriber more specifically with advice on the duration of combined antithrombotic therapy [18].

Therefore, the antithrombotic questionnaire tool can help community pharmacists to acquire the necessary information to perform adequate medication surveillance for their patients using combined antithrombotic therapy. However, for this new tool to be widely implemented in community pharmacies, it is important to gain insight into pharmacists’ and pharmacy technicians’ views on factors that might influence its use in daily practice.

The objective of this study was to identify potential barriers and facilitators that influence the implementation of the antithrombotic questionnaire tool in daily community pharmacy practice.

## 2. Materials and Methods

### 2.1. Study Design

This was a qualitative study in Dutch community pharmacies, based on semi-structured interviews with pharmacy staff who used the antithrombotic questionnaire tool. The study was assessed by the ethics committee of the Amsterdam University Medical Center, location VU University Medical Center, (2020.451) (Amsterdam, The Netherlands) and was not subject to the Medical Research Involving Human Subjects Act (WMO); therefore, it did not require a formal review. Patients and pharmacy staff provided informed consent prior to participation.

### 2.2. Setting

Dutch community pharmacies were contacted through the professional networks of the Pharmacy Foundation of Haarlem Hospitals and Nivel, the Netherlands Institute for Health Services Research. We intended to include fourteen pharmacies who would use the antithrombotic questionnaire tool with ten patients each. We expected that with this number of participating pharmacies would achieve data saturation. The literature suggests that around 12 interviews are sufficient to achieve data saturation, but that saturation also depends on nature, scope, and design of the study [19,20].

### 2.3. Study Population

In each participating community pharmacy, one or two team members were allocated to use the tool. The pharmacists and pharmacy technicians were both allowed to use the tool. An algorithm in the community pharmacy system was used to select patients for whom the antithrombotic tool could be used. Included combinations of combined antithrombotic therapy were DAPT, dual antithrombotic therapy (DAT), and triple antithrombotic therapy (TAT). DAPT consists of two platelet inhibitors, acetylsalicylic acid or carbasalate calcium, in combination with a P2Y12 inhibitor (clopidogrel, prasugrel, ticagrelor). Carbasalate calcium is a chelate of calcium acetylsalicylate and urea and is converted into acetylsalicylic acid in the gastrointestinal tract. DAT consists of a platelet inhibitor with an anticoagulant. TAT consists of two platelet inhibitors with one anticoagulant. Anticoagulants could either be vitamin K antagonists (VKAs), direct oral anticoagulants (DOACs), or low molecular weight heparins (LMWHs). Whilst this study focused on the implementation of the antithrombotic questionnaire tool, we also intended to assess the correctness of the antithrombotic therapy based on the patients’ answers to provide feedback to the pharmacists on possible incorrect antithrombotic therapy. The results of this assessment are presented in Appendix A. Therefore, patients were asked to provide written consent to have their anonymized data sent for assessment to the research team.

### 2.4. Interview Structure

The use of a framework helps to organize concepts and data without specifying causal relationships. The Consolidated Framework for Implementation Research (CFIR) is a widely used tool to identify barriers and facilitators in health care implementation research [20]. The CFIR consists of five domains and each domain consists of several constructs and subconstructs. From each domain, the constructs and subconstructs were selected that were most important for the implementation of this study (Table 1). The CFIR online interview tool developed by Damschroder et al. was used to formulate interview questions according to the constructs of the CFIR [21].

For each relevant construct, one or more interview questions were formulated. All questions were reviewed for relevance and completeness of the CFIR by the research team (M.A.B, R.C.A.E.v.U, M.V, M.L.B.). Consensus of the selection of constructs and questions was achieved after multiple meetings with the study team. A semi-structured interview guide was composed, which was divided into two sections, i.e., a general section and an experience and opinion section. Some questions were specific for a pharmacist or a pharmacy technician. Therefore, two interview guides were developed, i.e., one for pharmacists and one for pharmacy technicians. For instance, pharmacy technicians were asked what kind of support they received from the pharmacist before using the antithrombotic questionnaire tool. The complete interview guidelines for pharmacists and for pharmacy technicians can be found in Appendix A, respectively. The interviews were held in Dutch, the quotes of the interviews were translated in this paper to English.

### 2.5. Use of Antithrombotic Questionnaire Tool

Pharmacies were free to decide how they wanted to contact the patients and in what way they wanted to administer the tool. Pharmacies were instructed to use the same method for all patients within the pharmacy. The pharmacy team member could administer the antithrombotic questionnaire tool via telephone, video call, during pick-up of medication in the pharmacy, or during a medication review. The pharmacies used the tool from September until November 2020.

### 2.6. Data Collection and Interview Situation

Interviews were conducted by M.A.B. by video calls in November 2020. Age, sex, and work experience were assessed of each interviewee as well as the location of the pharmacy and size of the pharmacy team where the interviewee worked. The interviews were audio recorded and transcribed verbatim using Amberscript^®^ software (Amberscript BV Amsterdam, The Netherlands). M.A.B. performed a quality check to compare the transcripts to the original records for inconsistencies.

### 2.7. Data Analysis

The interview data were analysed using deductive thematic analysis. A coding tree was created before the analysis started, in which the selected CFIR domains represented the metacodes and the selected constructs within these domains represented the subcodes. New codes could be added when a text segment could not be assigned to one of the pre-specified codes. MAXQDA Analytics Pro^®^ 2020 version 20.3 (VERBI GmbH, Berlin, Germany) was used to perform coding. The researchers (M.A.B. and R.C.A.E.v.U.) individually coded all transcripts. Comparison of the codes revealed a high degree of consensus. A third researcher (M.V.) was consulted to discuss doubts in coding, which were resolved by discussion.

## 3. Results

### 3.1. Characteristics of the Pharmacies and the Pharmacy Staff

After 18 pharmacies were contacted, 14 pharmacies indicated their intent to participate, of which 10 pharmacies actually participated and used the antithrombotic questionnaire tool. The reasons to withdraw from participation were lack of time due to the COVID-19 pandemic (*n* = 2) and difficulties in obtaining informed consent (*n* = 2). During the study, several pharmacists indicated that they experienced difficulties with the inclusion of patients. Pharmacists indicated that patients were willing to answer questions on their antithrombotic therapy. However, due to the COVID-19 pandemic, most patients were contacted by phone and not all patients returned the signed informed consent form.

In seven pharmacies, the questionnaire was used by pharmacists, and in three pharmacies, the questionnaire was used by pharmacy technicians. One pharmacist participated with two pharmacies; therefore, nine interviews were conducted. The interviews took, on average, 30 min to conduct. The interviewed pharmacy team members were aged between 24 and 56. Other characteristics of the participating pharmacies are shown in Table 2. The questionnaire was completed for 82 patients (2–11 patients per pharmacy). One pharmacy technician used the questionnaire twice, the other pharmacy team members used it at least four times. In six pharmacies, the questionnaire was administrated by phone, and in three pharmacies, the questionnaire was administered partially by phone and partially face-to-face in the pharmacy. Data saturation was achieved. In the last interviews, no new themes emerged in the interviews.

### 3.2. Intervention Characteristics

#### 3.2.1. Relative Advantage

All pharmacists indicated that they already monitored the duration of combined antithrombotic therapy prior to the start of the study. The way in which this monitoring was performed differed between pharmacists. Most pharmacists performed a check on the intended duration of combined antithrombotic therapy at the initiation of therapy. This was mostly done by verifying whether the intended duration was written on the prescription. If the intended duration of combined antithrombotic therapy was not mentioned or unclear, most pharmacists indicated that they would call the prescribing physician to ask for the intended duration of the therapy. To monitor unintentional continuation of antithrombotic therapy, most pharmacists used an algorithm to identify patients who used combined antithrombotic therapy for over a year.

All pharmacists thought that even though they already checked the correctness of combined antithrombotic therapy that the questionnaire would be a good addition to regular medication surveillance. The questionnaire provided additional information, for instance, on the indication of antithrombotic therapy. The questionnaire was easy to use and with the information obtained from the questionnaire it took the pharmacist less time to assess whether the antithrombotic therapy was correct or possibly incorrect.


*“I see the list with alerts of the medication surveillance for duplicate medication and pseudo duplicate medication. And you see the prescriptions during the prescription check, so I make a note in the pharmacy information system. If it is not clear based on the prescriptions and I see an alert and it is not in the patient pharmacy record, then I will call the patient or I try the physician. So, I think with this questionnaire, if you can use it in daily practice at the moment the patient picks up his prescription from the pharmacy technician. This could be very useful.”*
(Pharmacist, male, 29 years old)

#### 3.2.2. Adaptability

Most pharmacists and pharmacy technicians indicated that no alterations in the content of the questionnaire were necessary in order for the questionnaire to be used effectively in their setting. Some mentioned that they would like to make small alterations in the layout of the questionnaire. One pharmacist indicated that the term “atrial fibrillation” was too complicated for patients and suggested to use a more simple term such as heart rhythm disorder. Another suggestion was given to add the dosage of the antithrombotics in the questionnaire. Some mentioned that, ideally, the questionnaire should be integrated in the pharmacy information system.


*“Like I said earlier, I think the questionnaire is clear and well drafted, so I would not suggest to adjust it.”*
(Pharmacy technician, male 35 years old)

All but one interviewed participant stated that they used the questionnaire as indicated and did not skip questions. One pharmacist stated he adjusted the questions to better fit the conversation with the patient. These adjustments were shorter sentences and word choices based on patients’ understanding.

#### 3.2.3. Complexity

All of the interviewed pharmacy team members found the patient questionnaire easy to use. The average time spent to use the questionnaire was ten minutes with a range of five to fifteen minutes. Participants stated that the duration varied per patient because answering the questions triggered some patients to ask more about their medication.


*“I do not think the questionnaire is complicated, no, I think that it is clear.”*
(Pharmacy technician, female, 35 years old)

### 3.3. Outer Setting

#### 3.3.1. Patient Needs and Resources

The pharmacists stated that most patients appreciated the attention of the pharmacy team concerning their medication and were willing to answer the questions. However, some patients had no interest in participating because they were afraid that their medication would change. Other patients did not want to take the effort to answer the questions. Pharmacists estimated that the questionnaire could be used for 70–80% of their patient population based on the patients’ health literacy skills.

#### 3.3.2. Cosmopolitanism

Some pharmacists stated that they believed that their contact with physicians could change when they would implement the questionnaire. The pharmacists stated that when they had more information about the indication of the antithrombotics that they were better equipped when calling the physician.


*“Well, I think you will be better prepared when you make a proposal to the physician. So that’s it mainly. Otherwise, you might discuss it with a physician’s receptionist first to gain some information about the history. Yeah, so that’s it mostly I think.”*
(Pharmacist, female, 27 years old)

### 3.4. Inner Setting

#### 3.4.1. Implementation Climate

All the pharmacies participate in various projects and in internships for pharmacy students. Some pharmacies stated that they participate in these projects because they think it is important to improve health care.

##### Compatibility

Views on the compatibility of the questionnaire in daily practice differed among the pharmacies. Some pharmacists wanted to use the questionnaire when dispensing a new prescription of combined antithrombotic therapy. The questionnaire would then be used by the pharmacy technician. Other pharmacists thought that it would take too much time during the pick-up of a prescription and that they would rather use the questionnaire a few weeks after the start of the antithrombotic therapy. Another pharmacist wanted to use the questionnaire during their check of patients who used combined antithrombotic therapy. On the one hand, some pharmacist had doubts whether (all) pharmacy technicians would be able to use the questionnaire, since they would not use this questionnaire frequently enough to become familiar with it. On the other hand, the three pharmacy technicians who used the questionnaire thought that the questionnaire could be used by pharmacy technicians. All pharmacists mentioned that a digital version of the questionnaire would make it easier to use.


*“I think from my own experience, that this cannot be used broadly by pharmacy technicians. I do not think so, because it does not happen that often. Look, we [pharmacists] see this questionnaire and then we use it. You read it once and then you can use it and indeed, for pharmacy technicians, they have to study it better. Get used to it, and would need more preparation time.”*
(Pharmacist, female, 30 years old)

##### Relative Priority

Possible reasons for not giving priority to the questionnaire varied. Some pharmacists thought a long waiting queue could be a possible barrier to use the questionnaire during the pick-up of medication. However, most pharmacists regarded the combined antithrombotic therapy to be a high-risk medication and prioritized it during medication surveillance regardless of the workload. One pharmacist stated that he already knew the intended stop date for the patients who were using combined antithrombotic therapy. However, he also stated that, when this information was not available for his patients, the questionnaire would be a good alternative.


*“Time pressure is not a barrier for me, because I always take care of the antithrombotic interventions, regardless of time pressure. So, the check in patients who pop up in the algorithm must be completed at least every two weeks. Period.”*
(Pharmacist, female, 30 years old)

#### 3.4.2. Readiness for Implementation

##### Leadership Engagement

The pharmacy technicians were asked what type of instruction or support they received from the pharmacist. All technicians stated that the pharmacist provided them with the information letter from the researchers with instructions on how to use the questionnaire and a list of patients that fit the inclusion criteria. The technicians mentioned that the pharmacist was available in case they had questions about (the use of) the questionnaire.

##### Available Resources

The pharmacy technicians were asked if they received enough time to use the questionnaire. Most technicians answered that they received enough time initially. However, due to the COVID-19 pandemic, pharmacy team members were more often absent due to illnesses, and the workload increased for the remaining employees. Therefore, less time was available for extra tasks, such as completing these questionnaires.

##### Access to Knowledge and Information

Pharmacy technicians were asked about how they dealt with uncertainties about the questionnaire and who they would approach in case they needed support. The technicians stated they would have approached the pharmacist; however, none of them had uncertainties during the study and they did not approach the pharmacist. The participating pharmacists were asked if the questionnaire had provided them with enough information to analyse the correctness of the patient’s therapy. No pharmacists stated that they needed more information.

### 3.5. Characteristics of Individuals

#### 3.5.1. Knowledge and Beliefs about the Intervention

Most participants were enthusiastic about the questionnaire. Some pharmacists and pharmacy technicians were amazed how much patients knew about their health condition. They thought that the questionnaire was a good addition to improve medication safety. Only one pharmacist stated he would not use the questionnaire after the study, because he already received the intended stop date in the prescriptions.


*“And then I found it quite interesting to see that they [patients] know for what [indication] they used it [medication] and for how long they should approximately use it. Because you doubt this as a pharmacy technician. Clearly, they know very well what is intended. Then they are well informed about that. I thought that was interesting.”*
(Pharmacy technician, female, 35 years old)

#### 3.5.2. Self-Efficacy

All interviewed participants, both the pharmacists and the pharmacy technicians, stated that they were confident to use the questionnaire.


*“Yes, I could use the questionnaire quite well. All questions were clear and were easy to use and could be filled in easily.”*
(Pharmacy technician, female, 27 years old)

#### 3.5.3. Individual Stage of Change

One pharmacy technician stated that he paid more attention to patients using combined antithrombotic therapy due to participation in this research. Most pharmacists intend to continue using the questionnaire. While the pharmacists could make the decision for themselves, the pharmacy technicians did not consult with their pharmacist to decide whether they wanted to continue using this questionnaire. One pharmacist already planned to assign a pharmacy technician with the task to conduct the questionnaire on a weekly base for all new patients with combined antithrombotic therapy.


*“I think that I will implement this, or a slightly adapted version. And that a specialized pharmacy technician will hold the conversations instead of the pharmacists.”*
(Pharmacist, female, 30 years old)


*“Yes, I would rather use this [the questionnaire] at the moment someone starts with combined antithrombotic therapy. So, the moment that the first prescription of combined therapy is received. So, if you would dream how it could be used ideally, then at the moment the pop-up [alert in the pharmacy information system] is shown, the questionnaire is linked to it and you can use it immediately. And then you can note [in the pharmacy information system] how long something [antithrombotic therapy] should be used and connect that with interventions.”*
(Pharmacist, female, 35 years old)

### 3.6. Process

#### 3.6.1. Planning

Most pharmacies contacted patients by phone. The questionnaire was conducted by phone in five pharmacies. Two pharmacies approached the patients during pick-up of medication or asked the patient to visit the pharmacy for conduction of the questionnaire. Two pharmacists stated they switched from in-person contact to telephone due to the COVID-19 restrictions.

#### 3.6.2. Barriers and Facilitators

Table 3 presents an overview of the (potential) barriers and facilitators for the implementation of the tool, extracted from the interviews. Most facilitators were found in the constructs complexity and compatibility. The main barriers were found in the constructs relative priority, available recourses, and self-efficacy.

#### 3.6.3. Suggestions for Optimization

Some pharmacists had suggestions for improvement of the questionnaire, for example a smoother flow through the questionnaire, subheadings could be placed, sentences could be shortened, and answer options should be clearer to select. For a better check of the medication, the dosage should be written down for each drug. One pharmacist is going to design a small card with information about antithrombotic drugs and will lecture the pharmacy team members about this subject. Based on the feedback from the users of the questionnaire an improved version of the questionnaire is included in Appendix A.

#### 3.6.4. Barriers Related to the Study

In addition to barriers for implementation of the questionnaire, participating pharmacies also experienced barriers related to the research. The most common barrier mentioned was obtaining written informed consent from patients. Reasons for the difficulties with obtaining informed consent were that patients did not understand the research or were afraid of what would happen with their data. In addition, the patient must actively sign the form and bring or send it back to the pharmacy. Therefore, not all pharmacies could include ten patients. The use of an informed consent is not necessary for daily practice, and therefore, this barrier was only applicable for the research setting and will not influence implementation.


*“In daily practice, it was sometimes hard to receive informed consent. So yes, that was the main barrier I experienced, especially with corona. A lot of people do not want to go outside, so yes, now I have tried to send them [informed consent forms] by mail and I hope that people will sent them back.”*
(Pharmacist, female, 27 years old)

Extra quotes are included in the Appendix A. 

## 4. Discussion

This study offers insight into the implementation of an antithrombotic questionnaire tool in community pharmacies to assess the correctness of combined antithrombotic therapy. This tool had already shown its effectiveness in a research setting. In this study, we investigated whether this successful intervention could be implemented in daily practice.

Most pharmacies provided positive feedback about using the questionnaire. All pharmacists thought that the questionnaire had added value by improving medication safety for their patients. Facilitators for implementation were that the questionnaire was easy to use and the relative short duration to administer the questionnaire. Another positive aspect was that the questionnaire could be used for a significant part of the patients with combined antithrombotic therapy. A possible barrier for using this questionnaire was a lower priority for using the questionnaire when the workload was high. Some pharmacist thought that the questionnaire might be too complex to be used by (all) pharmacy technicians, since combined antithrombotic therapy does not occur that often and they would not use the questionnaire frequently enough to become familiar with it. The questionnaire might not be suitable for all patients who use combined antithrombotics. Based on the suggestions made by the pharmacy team, the questionnaire has been optimized to enhance implementation (Appendix A).

Many interventions, for example, to improve medication adherence and medication safety have been developed for pharmacies [22,23,24]. However, the translation of these interventions into daily practice is a critical but often an ignored step [25]. There is a gap between research and practice because factors that are important for implementation are often not (well) investigated or identified alongside the evaluation of an intervention. Strategies for implementation are, therefore, based on best guess and not always based on systematic assessment of barriers and facilitators [26]. This emphasises the need to study the factors that influence implementation of our antithrombotic questionnaire tool.

Some studies have investigated facilitators that influence implementation of innovations in community pharmacies. Facilitators found in these studies were: enhanced contact with patients, improved workforce capability, and improved relationships with other healthcare providers. Professional recognition, personal or professional satisfaction, and integration of the intervention in the pharmacy information system were also identified as facilitators [27,28]. In our study, several similar facilitators were found such as enhanced contact with patients and improved relationships with other health care providers, improved professional satisfaction, and integration of the tool in the pharmacy information system. Barriers that have been mentioned in studies were the difficultly of the innovation, low public demand, and that patients were uninterested or reluctant [27]. Lack of time and increased workload, as well as lack of reimbursement and appropriate time were also mentioned as barriers [28,29]. Likewise, we found barriers such as patients who were not interested and lack of time in the case of an increased workload. Lack of reimbursement was not mentioned by pharmacists in our study. A reason for this could be that the number of patients who use combined antithrombotic therapy was relatively small, i.e., around 10 to 20 patients per pharmacy. Another reason could be that pharmacists believe that combined antithrombotic therapy is a high-risk therapy; and therefore, they consider that extra care for these patients is important to ensure medication safety.

A strength of this study is that we used the CFIR framework to structurally identify the barriers and facilitators. Additionally, this was a real-life setting to check if the questionnaire could be used in daily practice. A limitation of this study is that selection bias might be present. On the one hand, the selected pharmacies all participated voluntarily, and therefore, they could have been more motivated to implement new interventions and willing to make it successful. On the other hand, one could also argue that proactive pharmacists are more focused on medication surveillance, and therefore, would judge the questionnaire to be of less additional value. If the questionnaire is implemented by pharmacists who are less proactive, they might see an even higher additional value. Mostly due to the COVID-19 pandemic, we had only 10 pharmacies that participated, while our primary goal was to include 14 pharmacies. However, the number of included pharmacies was high enough since data saturation was obtained. One could argue that using the CFIR 2009 is a limitation of this study, since this framework was updated at the end of 2022 based on user feedback [30]; however, this update was published after completion of the present study. Moreover, the new CFIR framework is merely an extension of certain domains. Therefore, it is possible to map the new or adjusted constructs back to the original CFIR [30].

Based on the feedback from users, the questionnaire was adjusted for improved use in clinical practice. Ideally, the questionnaire should be integrated into the pharmacy information system so that the questionnaire pops up when a patient fills the prescription for combined antithrombotic therapy or that a pop-up is shown to the pharmacist at the moment of checking medication surveillance alerts. Not only the pharmacist, but also pharmacy technicians could play a crucial part in implementing the questionnaire in daily practice, as our study showed that pharmacy technicians were highly confident and faced no problems in using the tool. Pharmacists could use the questionnaire to expand their roles as health care professionals by providing extra focus on medication safety of this high-risk medication. Further research could focus on what patients think of the antithrombotic questionnaire tool.

Although pharmacists had concerns whether technicians were able to administer the questionnaire, the pharmacy technicians in our study were highly confident and faced no problems. However, it is worthwhile investigating whether the technicians remain confident in using the questionnaire over longer periods of time, and it might be beneficial for optimal implementation to designate one or two technicians per pharmacy for monitoring patients.

## 5. Conclusions

Our results indicate that the antithrombotic questionnaire tool can be implemented in daily pharmacy practice, as it is easy to use and is viewed by pharmacists as a valuable addition to regular medication surveillance. Attention should be paid to integrate this questionnaire in daily practice even when workload is high.

## Figures and Tables

**Table 1 pharmacy-11-00080-t001:** Selected domains and constructs of the Consolidated Framework for Implementation Research (CFIR) that were used for the interviews.

CFIR Domains	Constructs	Subconstructs
Intervention Characteristics	Relative Advantage	
	Adaptability	
	Complexity	

Outer Setting	Patient Needs & Resources	
	Cosmopolitanism	
Inner Setting		
Implementation Climate	CompatibilityRelative Priority
	Readiness for Implementation	Leadership EngagementAvailable ResourcesAccess to Knowledge & Information
Characteristics of Individuals		
Knowledge & Beliefs about the InterventionSelf-efficacy Individual Stage of Change
Process	Planning	


**Table 2 pharmacy-11-00080-t002:** Characteristics of the pharmacies.

Pharmacy	Tool Used by Pharmacist or Pharmacy Technician (Male/Female)	Located in Health Care Center	Phone/ Face to Face	Number of Pharmacy Technicians (FTE)	Working Experience in Years	Completed Questionnaires
A	Pharmacist (male)	No	Phone and face to face	4	1.5	10
B *	Pharmacist (female)	No	Phone	6.7	31	10
C	Pharmacy technicians (female)	Yes	Phone	6.5	8 and 2	7
D	Pharmacy technician (male)	No	Phone and face to face	3.6	3.5	10
E *	Pharmacist (female)	No	Phone	8	30	4
G	Pharmacy technician (female)	No	Phone	3.6	5	2
H	Pharmacist (female)	No	Phone and face to face	4.5	2.5	11
J	Pharmacist (female)	Yes	Phone	8.9	5	10
M	Pharmacist (female)	No	Phone	5	10	10
N	Pharmacist (male)	Yes	Phone	9	2	8

* In pharmacies B and E the antithrombotic questionnaire tool was used by the same pharmacist.

**Table 3 pharmacy-11-00080-t003:** Overview of barriers and facilitators presented per construct.

(Sub)Construct	Potential Barrier	Facilitator
Relative advantage		Addition to regular medication surveillance.
Adaptability		Easily adaptable
Complexity		Easy to use with patients. Small time investment
Patient needs and resources	The questionnaire might not be suitable for all patients.	The questionnaire could be used for around 70–80% of the patient population based on the patient’s health literacy skills.
Cosmopolitanism		Can improve contact with physician
Implementation climate		Receptive pharmacists who are open for innovative projects
Relative priority	Lower priority when workload is high	
Access to knowledge and information		Well understood instructions/information materials (of easy to use materials)
Knowledge and beliefs about the intervention		Addition to improve medication safety.
Self-efficacy	Questionnaire might be too complex for pharmacy technician (view of some pharmacists)	High confidence/easy to use

## Data Availability

The datasets used and/or analysed during the current study are available from the corresponding author on reasonable request.

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
