# Peer review of "Implementation of a Patient Questionnaire in Community Pharmacies to Improve Care for Patients Using Combined Antithrombotic Therapy: A Qualitative Study"

_pharmacy, 2023, doi:10.3390/pharmacy11030080_

Round 1
Reviewer 1 Report
Thank you for the opportunity to review this study manuscript. Overall, the structure and execution of the study is sound. They authors stayed within the study objectives and drew sound conclusions based on their results. At the apparent time of this study, the CFIR 2009 would be the most recent version of the available constructs. It may be worth mentioning in the paper that the CFIR was updated in 2022 but was obviously unavailable for constructing this study. This is relevant because some of the constructs used have changed. I do not think this impacts the relevance of this study or its consideration.
In each section, it may be of benefit to the reader to quantify the percentages associated with some of the statements. For example, in adaptability it is stated that "Most pharmacists...". This could be accompanied by a %. I don't think this is essential to publication, but just a suggestion.
Overall, I think the application of the CFID supports the design and the thematic analysis is well applied. I recommend the manuscript for publication with consideration of the above.
Author Response
Thank you for the opportunity to review this study manuscript. Overall, the structure and execution of the study is sound. They authors stayed within the study objectives and drew sound conclusions based on their results.
At the apparent time of this study, the CFIR 2009 would be the most recent version of the available constructs. It may be worth mentioning in the paper that the CFIR was updated in 2022 but was obviously unavailable for constructing this study. This is relevant because some of the constructs used have changed. I do not think this impacts the relevance of this study or its consideration.
Thank you for reviewing our manuscript. We are aware of the updated version of the CFIR, which was indeed done after we conducted our study. We agree that this update is worth mentioning in the manuscript. We added the following phrases to the discussion part of the manuscript:
“One could argue that using the CFIR 2009 is a limitation of this study, since this frame-work has been updated at the end of 2022 based on user feedback [30]. However, this up-date was published after completion of the present study. Moreover, the new CFIR frame-work is merely an extension on certain domains. It is therefore possible to map the new or adjusted constructs back to the original CFIR [30].“
In each section, it may be of benefit to the reader to quantify the percentages associated with some of the statements. For example, in adaptability it is stated that "Most pharmacists...". This could be accompanied by a %. I don't think this is essential to publication, but just a suggestion.
Overall, I think the application of the CFID supports the design and the thematic analysis is well applied. I recommend the manuscript for publication with consideration of the above.
We appreciate your suggestion. We believe that in a qualitative study quantification of the results may lead to misinterpretation of the results. When providing percentages, we may suggest a precision that is inappropriate because of the limited number of interviewed participants. Therefore we choose not to provide any percentages.

Reviewer 2 Report
This is a qualitative study involving pharmacists and pharmacy technicians in the community setting in the Netherlands regarding the utility of an antithrombotic assessment tool for patients on DAPT, DAT, or TAT therapies. The manuscript is well written and referenced. The title and abstract appropriately describe the study. Keywords are inclusive and MeSH terms. Appropriate software were used to transcribe interviews verbatim and to fracture and reconstruct the interviews. Conclusion follows from the results and the tool has practical utility in advancing pharmacist-patient interaction and monitoring within a hazardous therapeutic area. While the authors mention in the discussion that ideally the tool would be integrated into pharmacy computer systems, they make no statement about the practicality of storing collected data from the tool or how it would be stored and accessed.
My criticisms of the manuscript are three-fold: (1) the dearth of in vivo quotes from the interviews; (2) the lack of a conceptual framework or "big picture" that shows the relationship and connection between domains; and (3) lack of inclusion of all qualitative data in an appendix. First, only 10 quotes were used and in only one instance, two quotes were used to illustrate points. In some domains, there were no quotes used. Second, how the themes relate to each other is important to the reader and this display would add rigor to the presentation. Visualization of the data, either with word clouds or heat maps that MAXQA analytics could provide, would bring the data to life. Third, while optional, there is no indication of the volume of text transcribed from the interviews. MAXQA can also quantify qualitative data. However, the authors make the data available on request.
References are double numbered and not in MDPI style.
Thank you for the opportunity to review and comment on your interesting research.
Author Response
My criticisms of the manuscript are three-fold: (1) the dearth of in vivo quotes from the interviews; (2) the lack of a conceptual framework or "big picture" that shows the relationship and connection between domains; and (3) lack of inclusion of all qualitative data in an appendix. First, only 10 quotes were used and in only one instance, two quotes were used to illustrate points. In some domains, there were no quotes used. Second, how the themes relate to each other is important to the reader and this display would add rigor to the presentation. Visualization of the data, either with word clouds or heat maps that MAXQA analytics could provide, would bring the data to life. Third, while optional, there is no indication of the volume of text transcribed from the interviews. MAXQA can also quantify qualitative data. However, the authors make the data available on request.
- Depth of the in vivo quotes from the interviews and only 10 quotes were used.
Thank you for reviewing our manuscript. We added some quotes in order that all domains have quotes. Only the constructs patients needs and resources and planning have no quote. We added some extra quotes in a table in the new appendix 6. In most qualitative studies one or two quotes per paragraph are used. Lingard describes in the article; Beyond the default colon: Effective use of quotes in qualitative research, that more quotes are not necessarily better and that one quote should be sufficient to illustrate your point and that some points in your argument may not require a quoted excerpt at all. (Lingard Perspect Med Educ 2019 360-364). A Dutch Book on analysing in qualitative research by Boeije et al. 2019 advises to use 1 or 2 quotes in qualitative research setting. Therefore, we believe that with the addition of the quotes in appendix 6 sufficient quotes were used.
Appendix 6 Additional Quotes
|
(sub)Construct |
Quote |
|
Complexity |
“When you read the questionnaire well before you use it then it is not complex to use” (Pharmacy technician, female 24 years old) |
|
Cosmopolitism |
We had 3 similar quotes from pharmacy staff on this construct. “If you already know the indication and you can estimate the expected duration than you can call the physician more prepared.”
|
|
Compatibility |
“ You do need to take the time for the questionnaire, you can’t just do it in between jobs or on the counter of the pharmacy, therefore the conversation [use of questionnaire] is too long in a busy pharmacy” (pharmacy technicians, female age 24 and age 27)
|
|
Knowledge and beliefs about the intervention
|
“Yes, most of the patients know quite something (about their anticoagulation therapy) that was remarkable to me. So that was good to see. And then you see the added value of the questionnaire and that you can use it quite well in daily practice so you can better estimate what kind of anticoagulation someone needs. So I really liked using it [the questionnaire].” (Pharmacist, female, 27 years-old)
|
|
Relative priority |
“ Time pressure could be a barrier yes, when there is a line of patients that are waiting then you do not use the questionnaire. To use the questionnaire, you need to take the patients apart or make an appointment and in the normal workflow sometimes things can be forgotten” (Pharmacist, female, 56 years old)
|
|
Self-efficacy |
“I believe that pharmacy technicians could use this questionnaire when clearly is explained to them for which indications two or more antithrombotics are indicated. Then I believe that pharmacy technicians could use this questionnaire.” (Pharmacy technician, female, 35 years old)
|
2) The lack of a conceptual framework or "big picture" that shows the relationship and connection between domains; How the themes relate to each other is important to the reader and this display would add rigor to the presentation. Visualization of the data, either with word clouds or heat maps that MAXQA analytics could provide, would bring the data to life.
The CFIR consists of five major domains; interventions characteristics, outer setting, inner setting, characteristics of individuals and process of implementation. Each domain consists of several constructs and sometimes subconstructs. These domains interact in rich and complex ways to influence implementation effectiveness. All domains were used for this study. We chose the constructs and subconstructs that would reflect best how the questionnaire could be used in daily practice and to detect which barriers and facilitators could be involved.
Damschroder et al. describe that the domains of the CFIR framework interact in rich and complex ways to influence implementation effectiveness. (Damschroder et al. 2009 Implementation Science https://implementationscience.biomedcentral.com/articles/10.1186/1748-5908-4-50 )
We believe that by following the CFIR, the connections between domains are clear and we do not think that adding a word cloud would improve these connections.
We added some additional information in paragraph 2.4 interview structure. (additions in bold and italic)
The use of frameworks helps to organize concepts and data without specifying causal relationships. The Consolidated Framework for Implementation Research (CFIR) is a widely used tool to identify barriers and facilitators in health care implementation research [20].The CFIR consists of five domains (interventions characteristics, outer setting, inner setting, characteristics of individuals and process of implementation. “The domains interact with each other to influence implementation effectiveness.[20] All domains were used for this study. Each domain consists of several constructs and sometimes subconstructs.” From each domain the constructs and subconstructs were selected that were most important for this implementation study (Table 1). The CFIR online interview tool made by Damschroder et al. was used to formulate interview questions according to the constructs of the CFIR.
3) Third, while optional, there is no indication of the volume of text transcribed from the interviews. MAXQA can also quantify qualitative data. However, the authors make the data available on request.
The interviews took on average 30 minutes (lines 198-199). Software was used to transcribe the audio to text in a non-verbatim way (transcribing all spoken language but omitting background noises, verbal pauses, etc). This gives an indication of the volume of text transcribed. If readers are interested in the data, the data are available on request.
